# From Genetic Alterations to Tumor Microenvironment: The Ariadne’s String in Pancreatic Cancer

**DOI:** 10.3390/cells9020309

**Published:** 2020-01-28

**Authors:** Chiara Bazzichetto, Fabiana Conciatori, Claudio Luchini, Francesca Simionato, Raffaela Santoro, Vanja Vaccaro, Vincenzo Corbo, Italia Falcone, Gianluigi Ferretti, Francesco Cognetti, Davide Melisi, Aldo Scarpa, Ludovica Ciuffreda, Michele Milella

**Affiliations:** 1Medical Oncology 1, IRCCS Regina Elena National Cancer Institute, 00144 Rome, Italy; chiara.bazzichetto@ifo.gov.it (C.B.); vanja.vaccaro@ifo.gov.it (V.V.); italia.falcone@ifo.gov.it (I.F.); gianluigi.ferretti@ifo.gov.it (G.F.); francesco.cognetti@ifo.gov.it (F.C.); 2Department of Diagnostics and Public Health, Section of Pathology, University and Hospital Trust of Verona, 37134 Verona, Italy; claudio.luchini@univr.it; 3Division of Oncology, University of Verona, 37126 Verona, Italy; francesca.simionato@univr.it (F.S.); michele.milella@univr.it (M.M.); 4Medicine-Digestive Molecular Clinical Oncology Research Unit, University of Verona, 37126 Verona, Italy; raffaelasantoro@gmail.com (R.S.); davide.melisi@univr.it (D.M.); 5ARC-Net Research Centre, University and Hospital Trust of Verona, 37126 Verona, Italy; vincenzo.corbo@univr.it (V.C.); aldo.scarpa@univr.it (A.S.); 6SAFU, Department of Research, Advanced Diagnostics, and Technological Innovation, IRCCS Regina Elena National Cancer Institute, 00144 Rome, Italy; ludovica.ciuffreda@ifo.gov.it

**Keywords:** pancreatic cancer, oncogene, tumor suppressor, signaling pathway, tumor microenvironment, targeted therapy

## Abstract

The threatening notoriety of pancreatic cancer mainly arises from its negligible early diagnosis, highly aggressive progression, failure of conventional therapeutic options and consequent very poor prognosis. The most important driver genes of pancreatic cancer are the oncogene *KRAS* and the tumor suppressors *TP53*, *CDKN2A,* and *SMAD4*. Although the presence of few drivers, several signaling pathways are involved in the oncogenesis of this cancer type, some of them with promising targets for precision oncology. Pancreatic cancer is recognized as one of immunosuppressive phenotype cancer: it is characterized by a fibrotic-desmoplastic stroma, in which there is an intensive cross-talk between several cellular (e.g., fibroblasts, myeloid cells, lymphocytes, endothelial, and myeloid cells) and acellular (collagen, fibronectin, and soluble factors) components. In this review; we aim to describe the current knowledge of the genetic/biological landscape of pancreatic cancer and the composition of its tumor microenvironment; in order to better direct in the intrinsic labyrinth of this complex tumor type. Indeed; disentangling the genetic and molecular characteristics of cancer cells and the environment in which they evolve may represent the crucial step towards more effective therapeutic strategies

## 1. Introduction

Pancreatic cancer (PC) is perhaps the most lethal among solid tumors worldwide; although incidence varies between different countries, PC is the seventh leading cause of global cancer-related deaths in industrialized countries, the fourth in Italy, and the third in the United States of America [1,2,3]. Long-term survival is poor, with a five-year survival rate ranging from 5% to 8% [3]. The threatening notoriety of PC primarily arises from the lack of sensitive and specific biomarker(s) for detection and its negligible early diagnosis: indeed, the vast majority of PC patients are not eligible for surgical resection at the time of the diagnosis and present with a locally advanced or metastatic stage [4]. Moreover, even in patients undergoing potentially curative surgical resection the disease usually recurs within 2–3 years in the majority of cases.

As highlighted above, such PC unique aggressiveness is due to both specific genetic and molecular traits (acquired through either a stepwise or a chromothripsis-mediated simultaneous loss of multiple tumor suppressors and gain of multiple oncogenes), but to complex interactions with a generally obstructive and immunosuppressive tumor microenvironment (TME), which confer a highly malignant phenotype, as well (Figure 1) [5,6].

Although combination chemotherapy has to some extent improved cure rates, survival, and quality of life in both early and advanced PC, anti-angiogenic drugs, molecularly targeted agents and immune checkpoint inhibitors have proven ineffective, so far, in PC [7,8]. Even the combination of therapies simultaneously targeting multiple crosstalking pathways has failed to confer clinical benefit in the absence of predictive biomarkers able to identify patients who will most likely respond to specific therapeutic combinations. This is well exemplified by the recent failure of a theoretically rational combination of MEK and AKT inhibitors [9]. In that respect, our group has recently identified PTEN-loss status as a predictive biomarker of synergistic pharmacologic interactions between mitogen-activated protein kinase (MAPK)- and phosphoinositide-3-kinase (PI3K)-inhibitors, in several preclinical models in vitro; when applied to PC the combination of data obtained in vitro and the relative rarity of complete loss of PTEN function in pancreatic ductal adenocarcinoma (PDAC) could have predicted the clinical failure of the MEK/AKT combination in unselected PC patients [9,10].

In recent years the genetic pathway to PC development (encompassing a relatively restricted set of recurrently altered genes, such as *KRAS* (Kirsten rat sarcoma), *TP53* (tumor suppressor protein 53), *CDKN2A* (cyclin dependent kinase inhibitor 2A), and *SMAD4* (SMAD family member 4), and molecular pathways) has been elucidated to an unprecedented extent; on the other hand, it is now clear that an intense crosstalk between stroma and cancer cells, mediated by soluble factors, leads to extensive fibrosis and deposition of extracellular matrix (ECM) proteins, resulting in poor perfusion and hypoxic conditions which, in turn, favor the evolution of aggressive clones (Figure 1). The molecular mechanisms of such interactions are only beginning to be elucidated and the Ariadne’s string linking specific tumor genetic/molecular background(s), to the production of specific sets of soluble factors and to the formation of an obstructive and immune suppressive TME will need to be precisely identified to develop new and effective therapeutic strategies to defeat such an aggressive and therapy resistant disease.

## 2. Driver Genes Alterations and Molecular Pathways to PC Development

### 2.1. Precursor Lesions

The high-aggressive PC represents a late event in a time-manner dependent sequence of genetic and molecular events, such as pancreatic intraductal neoplasia (PanIN) and intraductal papillary mucinous neoplasm (IPMN). PanIN is the most common PC precursor. It is a microscopic (<0.5 cm) intraductal lesion that can be found >80% of pancreas with invasive carcinoma [11,12]. PanIN is composed by cuboid to columnar mucinous cells; the new World Health Organization classification distinguishes low- from high-grade dysplasia to classify possible varying degrees of dysplasia [12]. Seminal papers on this topic showed molecular evidences of the progression from PanIN to PC, with early lesions (low-grade PanINs) displaying *KRAS* somatic mutations [13,14,15]. In PanIN carcinogenetic cascade, the *TP53* and *SMAD4* inactivations appear as very late events, often exclusive of an already existing invasion [16]. Another important PC precursor in certainly represented by IPMN. IPMN is a grossly-visible lesion (>1 cm by definition), with intraductal growth and papillary architecture, composed of mucinous cells. IPMN dysplasia also should be classified in low- and high-grade [12]. Based on the involvement in pancreatic ductal tree, IPMN could be categorized in: (1) main-duct IPMN (involvement of only Wirsung’s duct), (2) branch-duct IPMN (involvement of only secondary ducts), (3) mixed IPMN (contemporary involvement of the main and the branch ducts). This classification displays very important implications in clinical practice, indeed the main-duct IPMN shows higher risks towards evolution in PC, as compared to the others two [12,17]. From a histological point of view, IPMN can be classified into four subgroups: gastric, pancreatobiliary, intestinal, and oncocytic [12]. Even this classification shows a clinical impact, due to the association of the pancreatobiliary subtype with PC development [18,19]. From a molecular point of view, the most frequently mutated genes in IPMN are *GNAS* (guanine nucleotide binding protein, alpha stimulating) and *KRAS*, which are altered in up to 60% and to 80% of cases, respectively [12,20,21]. Notably, recent researches highlighted that the IPMN oncogenetic process is regulated by two different pathways: the first, linked to *GNAS* mutations, involves intestinal IPMN progressing to colloid adenocarcinomas (a PC variant reach in extracellular mucin), and the second, driven by *KRAS* mutations, is typical of pancreatobiliary IPMN and leads to conventional PC [20,21].

### 2.2. Driver Genes Alterations

Our knowledge of the molecular bases of PC has recently greatly improved, owing to advances in technology (next-generation sequencing—NGS) and consortia-based approaches, the latter enabling the collection of large cohorts of carefully annotated specimens. From a genetic point of view, PC appears as a complex disease, with a number of genes being altered through different mechanisms including point mutations, chromosomal aberrations, and epigenetic mechanisms, resulting in an “intermediate” tumor mutational burden [22]. Four genes, also called “PC genetic mountains”, are most commonly mutated: the *KRAS* oncogene, the *TP53*, the *CDKN2A*, and the *SMAD4* tumor suppressor gene (Figure 1). Other genes altered at a lower but not-negligible prevalence are also called “PC genetic hills” [23,24]. Notably, alterations affecting the most important genetic drivers of PC can be demonstrated on tissue samples as well as by liquid biopsy, with reliable sensitivity and specificity [25].

#### 2.2.1. KRAS

The *KRAS* is an oncogene located on chromosome 12, and is the most frequently mutated gene in PC (>90% of cases); the vast majority of activating mutations occurs at codons 12, 13, or 61 [23,24,25,26,27]. This oncogene encodes a small GTPase, that is switched on and off by cycling between the GTP-bound (active) and GDP-bound (inactive) forms. It acts as a transducer-moderator, interacting with cell surface receptors (receptor tyrosine kinases); once triggered, it stimulates several intracellular effector pathways, which drive very important modifications of cancer cells, such as increased proliferation, metabolism and migration, immune system evasion, and apoptosis blockade [28]. These downstream pathways include, among others, MAPK and PI3K. Notably, attempts at inhibiting the activity of the mutant KRAS oncoprotein have been unsuccessful, thus rendering such target undruggable so far and, perhaps, one of the most important reasons for PC’s high-mortality rate [29]. Very recently, new data on KRAS G12C-selective inhibitors and other new-generation KRAS-targeted approaches have been reported, raising renewed hope that we could one day target the main genetic PC driver directly or through repurposed drugs [30]. Mutations affecting *KRAS* represent an early event in PC carcinogenesis, as they are found also in precursor lesions with low-grade dysplasia, such as PanIN and IPMN with low-grade dysplasia [31].

#### 2.2.2. TP53

The tumor suppressor gene *TP53* is located on chromosome 17 and is mutated in a significant percentage of PC cases, ranging from 50–60% to 80% [32,33,34,35,36]. This so-called “master gene” encodes for P53, a crucial protein that controls cell growth, metabolism, senescence, DNA repair, and apoptosis in response to different types of cellular and extracellular processes, such as hypoxia and DNA damage [37]. The most common alterations affecting *TP53* are represented by missense mutations and base substitutions in the DNA-binding domain, coupled with loss of the wild-type allele. Similar to *KRAS* mutations, *TP53* alterations are currently considered “undruggable” from a therapeutic perspective, thus representing another reasonable explanation for PC’s high-mortality rate. Mutations affecting *TP53* represent a late event in the PC carcinogenesis: indeed, they have been found in PanIN and IPMN with high-grade dysplasia or in the infiltrating component of PC only [31].

#### 2.2.3. CDKN2A

*CDKN2A* is a tumor suppressor gene located on chromosome 9. It is altered in a several different types of tumor, and its inactivation is reported in up to 95% of PC variants. The inactivation of this tumor suppressor can occur following three different processes: (1) intragenic mutations associated with loss of the second allele (40%); (2) homozygous deletion (40%); (3) promoter hypermethylation associated with the loss of the second allele (in remaining cases) [38,39]. P16, the protein encoded by this gene, acts as a mediator of the “retinoblastoma” (RB) signaling pathway. By inhibiting the phosphorylation of RB, P16 stops the entry of cells into the S phase of the cell cycle. Alterations of this signaling pathway thus promotes an uncontrolled cell growth and neoplastic progression [40]. Of importance, fluorescence in situ hybridization is a common method for evaluating homozygous deletion of the *CDKN2A* locus [41]. Alterations affecting *CDKN2A* represent a late event in the oncogenesis of PC [31].

#### 2.2.4. SMAD4

Alterations of the tumor suppressor gene *SMAD4* (also called *DPC4*), located on chromosome 18, are reported in a variable percentage (30% to 60%) [34,35,36,42,43,44]. This gene is inactivated by homozygous deletion or intragenic mutations matched with loss of the second allele. The protein encoded by *SMAD4* is an effector of the transforming growth factor (TGF)-β signaling pathway. In normal tissues, this pathway is important in preserving physiological homeostasis; indeed, it plays a crucial function in the control of cell proliferation. SMAD4 mutations and consequent perturbations of this pathway leads to a proliferative effect and by the activation of the process of the epithelial-to-mesenchymal transition (EMT) [45,46]. In this process, cancer cells lose their epithelial features and acquire a “mesenchymal” phenotype, which is essential for vascular invasion and overall metastatization. Recent studies have demonstrated that *SMAD4* mutations increase the fitness of PC cells to colonize distant organs, thus characterizing PC cases with widespread metastatization, and are usually absent in cases in which the disease remains locally aggressive for a relatively long time during its course [47]. Mutations affecting *SMAD4* represent a late event in the oncogenesis of PC: they have been found rarely in PanIN and IPMN with high-grade dysplasia, most commonly characterizing only the infiltrating component [31].

#### 2.2.5. Other Mutations

Among less commonly mutated genes, some, such as AT-rich interaction domain 1A (*ARID1A*) and TGF-β receptor (R) I, have also been reported in different studies and are defined as “genetic hills” in the PC landscape. The most important “hill” is represented by the *ARID1A* gene, located on chromosome 3. It is a tumor suppressor gene and its mutations are usually associated with poor prognosis in several types of solid malignancies [34,47]. It encodes for a component of the of SWI/SNF (BAF) complex, a multi-protein complex with important functions in chromatin remodeling. This is an epigenetic modulator, which can regulate the gene expression without modifying the DNA sequence [48]. Mutations affecting ARID1A can result in a profound dysregulation of the chromatin remodeling mechanism, with important advantages for the proliferation of cancer cells [48,49].

PC molecular landscape is also characterized by complex chromosomal alterations. A recent whole-genome sequencing based study, conducted by the International Cancer Genome Consortium, has found 11,868 somatic structural variants at an average of 119 variants per PC case [36]. In this study, PC has been classified into 4 different subtypes based on structural variations: (1) stable; (2) locally rearranged; (3) scattered; (4) unstable. Particularly, in the “unstable” subgroup of PC (characterized by marked genomic instability), tumors showed a significantly increased proportion of breast cancer gene 1 (*BRCA1*), *BRCA2*, and partner and localizer of BRCA2 (*PALB2*) mutations. As an immediate consequence, patients who received platinum-based therapy experienced very significant clinical benefit. At the same time, this study first highlighted the therapeutic potential of PARP-inhibiting agents, also in PC: this class of drugs, indeed, has recently emerged as a potential targeted therapy in selected, genetically characterized, cases [49]. In terms of targeted therapy, the *BRAF* (v-raf murine sarcoma viral oncogene homolog B1) oncogene may also be mutated in PC, but its prevalence is less than 2% of all cases [50]. A recent study has shown that PC in older patients (>50 years) were significantly more likely to harbor genomic alterations in *KRAS* and *SMAD4*, whereas *BRCA1* and *BRCA2* alterations were more commonly detected in PC from younger patients (<50 years). A higher frequency of genomic alterations in *KRAS*, *SMAD4* and *CDKN2A* were found in samples from female patients, whereas PC in male patients more often exhibited alterations in *GNAS* [50].

Some variants of PC also exhibit peculiar genetic alterations, not commonly observed in conventional PDAC. For example, the medullary PC variant often shows microsatellite instability (MSI), which can be targeted with immunotherapy [50,51,52,53,54]. The colloid PC variant, usually arising from IPMN, also has an increased rate of MSI and mutations in *GNAS*, located on chromosome 20, are also common in this variant [21,55,56]. Lastly, the PC variant with osteoclast-like giant cells is enriched for mutations in the *SERPINA3* oncogene, although its peculiar morphological aspect has been mainly linked to its specific inflammatory background rather than its genetic profile, which is quite overlapping to conventional PC [57].

### 2.3. Dysregulation of Core Signaling Pathways

Thirteen main individual signaling pathways are currently reported altered in PC (Table 1) [34,50,58].

The first NGS study conducted by Jones and colleagues analyzed a cohort of 24 PC, finding an average of 63 genetic alterations per case [34]. This study showed that the most common mutations were base substitution, mainly involving the four PC driver gene *KRAS*, *TP53*, *CDKN2A* and *SMAD4*. A significant intra-tumor heterogeneity was also described. Of note, the most important result of this pioneering investigation was represented by the finding that the vast majority of mutated genes belonged to 12 so-called “core-signaling pathways”. Such pathways, integrating also the results of more recent publications, can be summarized as follows: (1) the MAPK signaling pathway, which is regulated by KRAS GTPase and still remains the most difficult target for tailored treatments; (2) the DNA damage control, which includes TP53 and represents a crucial event for the acquisition of the invasive capacities by PC, being a late event during PC carcinogenesis; (3) the regulatory pathway of G1/S cell phase transition, essential for the preparation of DNA synthesis, with the participation of P16; (4) the TGF-β signaling pathway, which is mainly regulated by the *SMAD* family of genes; (5) the pathway of apoptosis, in which the caspase proteins play a very important role and are of importance for the proliferation of cancer cells; (6) the hedgehog pathway, recently emerged as a potential target for precision oncology, which is mainly regulated by *GLI3* gene products in PC; (7) the pathway of homophilic cell adhesion, in which the class of *CDH* and *FAT* genes are the most important effectors; (8) the pathway of integrin signaling, in which the encoded proteins regulate cell-to-cell adhesion and drive the process of local invasion of cancer cells; (9) the c-Jun N-terminal kinase (JNK) signaling, which is based on the energy transfer of phosphoryl groups and mainly involves *MAP4K3* and tumor necrosis factor (*TNF*) as most important genes; (10) the pathway belonging to chromatin remodeler-complexes (SWI/SNF), in which *ARID1A* is the most commonly mutated gene; (11) the signaling pathway dependent on small GTPases (KRAS excluded); (12) the Wnt/Notch signaling pathway, fundamental also in other types of cancer and represented by the *MYC* and *GATA6* genes as the main regulators; (13) to these pathways, another study has suggested to add a thirteenth, represented by the path of the embryonic driving regulators of the axon, which sees as the main signaling pathway the “so-called” SLIT/ROBO pathway [24,59]. Starting from the manuscript of Jones and colleagues, all the findings regarding the most important molecular pathway in PC have reinforced the vision of PC as a very complex disease with several different interactions and with a remarkable intra-tumor heterogeneity. Tailored therapeutic strategies in PC, based on a single gene only, have also for these reasons limited success for this patient setting [24]. Very recently, some clinical trials have initiated new approaches against PC, specifically using signaling pathway inhibitors (Table 2) [58].

Although the majority of these trials are of phase I, they represent one of the most important promising strategies of precision medicine for PC.

### 2.4. Molecularly-Defined PC Subtypes

Of interest is also reporting a recent attempt of classification of PC based on the frequency and distribution of structural rearrangements. Using data from whole-genome sequencing, Waddell and colleagues identified four different PC subtypes [36]. Subtype 1 was classified as “stable” and represented about 1/5 of cases. “Stable” was used to indicate that such neoplasms contained ≤50 structural variation events. Furthermore, this PC class often presented widespread aneuploidy suggesting defects in cell cycle/mitosis. Mutations of major PC driver genes were similar to the rest of the cohort. Subtype 2 was classified as “locally rearranged” and represented about 1/3 of all samples. This subtype presented a significant focal event on one or two chromosomes. About 1/3 of locally rearranged genomes contained regions of copy number gain that harbored known oncogenes, including focal amplifications with therapeutic targets (e.g., ERBB2). Another important molecular process discovered in this PC class was represented by chromothripsis. This process has been more recently described also in pancreatic neuroendocrine tumors, and causes fragmentation of a chromosome in many segments with consequent inaccurate reparations [60]. Subtype 3 was called as “scattered” and comprises, similarly to the “locally rearranged” class, about 1/3 of all cases. This PC subgroup presented a moderate range of non-random chromosomal damage and less than 200 structural variation events. At last, subtype 4 was called “unstable” and included about 15% of all samples. PC of this class harbored a large number of structural variation events (>200). This significant genomic instability suggested defects in DNA maintenance, with potential sensitivity to DNA-damaging agent. Of note, few patients within this group and with mutations affecting *BRCA* genes demonstrated clinical benefits from platinum-based therapy.

Several classifications have been also proposed based on mRNA expression analyses of bulk or microdissected tissues. Overall, when considering the characteristics of neoplastic cells all three classifications agree on the existence of two major lineages (classical/pancreatic progenitor and quasimesenchymal/basal-like/squamous) specified by different transcriptional programs and showing different prognosis and drug sensitivity [61,62,63]. In particular, the squamous/basal-like/quasi-mesenchymal lineage is characterized by loss of endodermal origin and activation of p63 and Dnp63 transcriptional programs. The analysis of bulk tissues also evidenced the existence of additional subtypes (Immunogenic and ADEX) which can be considered as classical/pp tumors but heavily infiltrated by non-neoplastic components, particularly leukocytes. Several studies have shown that a better understanding of the cell autonomous and non-cell autonomous determinants of molecular subtypes might lead to the design of better therapeutic intervention [64,65]. Along the same line, transcriptional rather than genetic signatures are being proven predictive of response to chemotherapy regimen or targeted therapy approaches [66].

## 3. TME

PC has long been described by pathologists as “a hypertrophic scar with sparse neoplastic cells”and stromal components may represent up to 85% of the tumor mass in PC, probably the most “desmoplastic” malignancy, among all solid tumors. PC desmoplastic stroma is a complex environment composed by both cellular and acellular components, such as cancer associated fibroblasts (CAF), immune cells (macrophages, neutrophils, myeloid-derived suppressor cells (MDSC), lymphocytes), and endothelial cells, on one hand, and collagen, fibronectin, and soluble factors (cytokines and chemokines), on the other (Figure 1). Such distinctive desmoplastic reaction around the tumor tissue is due to an intense autocrine and paracrine cross-talk between cancer and non-cancer components. Whether (or, more precisely, under what circumstances) TME represents a friend or a foe in terms of biological and clinical aggressiveness and response to therapy in PC remains a hotly debated topic. Indeed, TME elements with putative protumor properties are summarized in Table 3. Given the central role of TME in PC, understanding the biology behind tumor-stroma interactions (TSI) will be crucial to develop new therapeutic strategies for PC patients.

### 3.1. Cellular Components

#### 3.1.1. Stromal Cells

##### PSC and CAF

Quiescent Pancreatic Stellate Cells (qPSC) are involved in maintenance of pancreatic tissue architecture and lipid and vitamin A storage; in response to external stimuli (known PC risk factors (e.g., ethanol and smoking), environmental stress (e.g., hypoperfusion and hypoxia), soluble cellular factors (e.g., interleukin (IL)-6 and TGF-β) and activation of molecular signaling pathways (e.g., Wnt/β-catenin and PI3K)), they skew towards activated PSC (aPSC), acquiring alpha-smooth muscle actin (α-SMA) expression and enhancing ECM deposition and the release of soluble factors involved in angiogenesis and EMT. In PC, aPSC represent the most important source of CAF: they represent the predominant fibroblast cell type in PC stroma and share similar features, including increased expression of the most important myofibroblastic marker α-SMA and of the CXCL12 cytokine involved in lymphocyte recruitment [67,70,71,72,109]. For these reasons, it is quite accepted that aPSC and CAF represent the same cellular population, even though their classification is still debated and some groups maintain a distinction between the two cell populations [110,111].

Whether CAF and aPSC exert pro- or anti-tumor activity in PC remains a matter of debate, owing to opposite lines of experimental evidence. In general, both CAF and aPSC are involved in tumor proliferation and growth, metastasis and immunity disruption, thereby causing chemoresistance (Figure 1) [73,112]. For example, Wei and colleagues demonstrated that CAF release CXCL12 which in turn upregulates the expression of the nuclear matrix attachment region-binding protein SATB-1 in PC cells; through a reciprocal feedback loop between fibroblasts and tumor cells, SATB-1 is involved in maintenance of CAF features (i.e., α-SMA, fibroblast activation protein and CXCL12) and gemcitabine resistance [113]. Chemoresistance is also induced by hypoxic conditions, as revealed by multiple studies in which the authors demonstrated that stroma depletion upregulates intratumor perfusion and gemcitabine accumulation at the tumor site, thus promoting cytotoxicity and prolonging survival [114]. Another mechanism involved in gemcitabine-resistance is linked to the decrease of aPSC-mediated apoptosis, as demonstrated by Liu and coworkers. In the same paper, the authors also demonstrated the key role of another soluble factor, TGF-β1, in TSI: indeed, the TGF-β1 inhibitor SB525334 partially blocks aPSC and tumor cells viability and aPSC-dependent invasion [115]. A further signaling pathway involved in affecting the interactions between PSC and pancreatic tumor cells is the MAPK cascade, as very recently demonstrated by Amrutkar and collaborators. Indeed, activation of ERK1/2 in PC cells, mediated by fibronectin aPSC-released, is involved in gemcitabine resistance: consistently, inhibition of aPSC-released fibronectin or ERK signaling may counteract chemoresistance and deserves further investigation as a potential therapeutic strategy [69]. In clinical PC series, CAF activation appears to correlate with reduced OS: indeed, produced by CAF promote tumor cell invasion and a high proportion of CAF in primary tumors correlates with nodal and distant metastases and a high lymph node ratio [116].

On the other hand, in other reports low levels of α-SMA^+^ myofibroblasts correlated with poor survival of PC patients, thereby leading to the hypothesis that fibrosis constitutes a “protective” TME, rather than a tumor promoter. Alpha-SMA^+^ myofibroblasts knock-out mice display multiple adverse outcomes, such as reduced immunosurveillance and gemcitabine resistance; chemoresistance could be reverted by a combination of gemcitabine plus anti-CTLA-4 treatment, which resulted in increased animal survival [117].

##### Endothelial Cells

Another important element in PC stroma is represented by endothelial cells and the mutual influence between endothelial cells and PC microenvironment is now well established. Indeed, the mere presence of conditioned medium derived from hypoxic PSC increases angiogenesis by inducing endothelial cells proliferation and migration in vitro and in vivo [118]. Conversely, conditioned medium derived from PC cells reduces endothelial cells proliferation [119]. However, the reciprocal influence between these cells on their growth is not yet defined: indeed, Craven and her collaborators showed that PC cells promote endothelial cells expansion. This group has recently demonstrated that the concomitant inhibition of TGF-β1R and janus kinase (JAK) 2 suppresses the mitogenic effects exerted by both endothelial and PC cells [120]. The endothelial cells surface marker, CD31, is significantly associated with better prognosis and improved overall survival (OS) in PC patients. Indeed, tumors, which express high levels of CD31, are associated with higher infiltration of anti-cancer immune cells (e.g., CD8^+^ and CD4^+^ T cells) [74]. Moreover, CD31 correlates with SMAD4 expression. Therefore, it is not surprising that SMAD4, frequently mutated in PC tissue, is associated with the number of endothelial cells in PC [120].

#### 3.1.2. Immune Cells

In PC, cellular stromal components are also characterized by different immune cells which influence each other with endothelial cells to define an immunosuppressive TME.

##### Macrophages

Macrophages infiltrate PC attracted by vascular endothelial growth factor (VEGF)/VEGFR2 interactions [121] and affect its biological properties and clinical behavior. When polarized towards an M1 phenotype (HLA-DR^+^ and CD11c^+^), they exhibit pro-inflammatory properties; conversely, M2-polarized macrophages express CD163 and CD204 and exert tumor-promoting functions [108,122]. In PC TME hypoxic conditions and aPSC-derived cytokines, such as IL-10 and IL-13, polarize macrophages towards a M2, pro-tumor phenotype [123,124]. In addition, the expression of the toll-like receptor 4 on M2 macrophages promotes a vicious circle, stimulating further IL-10 release by tumor cells and promoting EMT through E-cadherin inhibition and Snail and vimentin upregulation [75]. In line with preclinical evidence, clinical data also confirm a dominant role of these cell populations within the PC immune landscape: indeed, macrophages are more represented in cancer tissue as compared to healthy pancreas, and their accumulation starts at the level of early stage lesions (PanIN) and increases during invasive cancer transformation [125]. In particular, macrophages exert an active role in conferring an aggressive behavior to primary PC through the regulation of the TGF-β [126]. Consistently, M2 populations represent a negative prognostic factor [76].

##### Neutrophils

Similar to macrophages, neutrophils represent another group of myeloid-derived circulating cells belonging to innate immunity. Despite their role in TME is still debate, the increasing knowledge of this cell type led to differentiate tumor associated neutrophils in N1 (with anti-tumor properties) and N2 (with pro-tumor properties), according with the type of cytotoxicity exerted and the ability to activate different immune cells [127]. Particular interest is actually attributed to neutrophil-to-lymphocyte ratio (NLR) as a prognostic biomarker in PC. In a recent metanalysis, Zhou and collaborators reported that patients with low NLR had significant smaller tumor size, longer OS and disease-free survival (DFS) as compared to high NLR patients [128]. Furthermore, the involvement of neutrophils in chemoresistance was recently investigated. Indeed, the combined blockade of CXCR2^+^ neutrophils and CCR2^+^ macrophages results in an increased antitumor immune response and response to the chemotherapy treatment with the FOLFIRINOX regimen, due to the reduction of infiltrating myeloid cells in the tumor [77].

##### Lymphocytes

Among immune cells, T lymphocytes are the most represented population and significantly influences the clinical outcome of PC patients [129]. According to the expression of specific surface markers, CD3^+^ T cells are classified in T helper CD4^+^, Treg CD4^+^/CD25^+^/Forkhead box P3 (Foxp3, and cytotoxic CD8^+^ [130]. CD4^+^ T cells mainly activate innate and acquired immunity, thus it is not surprising that their levels are lower in tumor tissue and peripheral blood of PC patients, as compared to healthy subjects; consistently, their infiltration is correlated with better OS [131,132]. On the other hand, in PC pro-tumor Th2 CD4^+^ (GATA3^+^) sub-populations are more represented than the tumor-killing Th1 (T-bet^+^) and their levels correlate with poor prognosis [133,134]. Tassi and her group indeed showed a potential benefit of immune reprogramming strategies aimed at reverting the predominant Th2 into a Th1 T cell phenotype preclinically, although clinical evidence of potential effectiveness is currently lacking [135]. Another subset of CD4^+^ cells characterized by the production of IL-17 (Th17) has been shown to play a pro-tumor role in PC, where both tumor-infiltrating Th17 CD4^+^ cells and circulating levels of Il-17 and IL-23 correlated with advanced stage and worse OS [78]. The well characterized Treg sub-population of immune cells contribute to an immunosuppressive TME by promoting CTLA-4 expression and TGF-β release [79,136]. In PC stroma the Treg population profoundly suppresses the activity of antitumor immune cells (i.e., CD4^+^, CD8^+^, macrophages and dendritic cells—DC) [79]. According to their tumor-promoting role, high levels of Treg cells are detected at the PC tumor site and circulating in the bloodstream, as compared to control subjects [137,138]. However, recent new data suggest that through their interactions with fibroblasts in PC TME, Tregs may actually counteract tumor development: indeed, it has been demonstrated that Treg depletion promotes pancreatic carcinogenesis, as a result of CAF activation and immunosuppressive myeloid infiltration [139].

CD8^+^ are cytotoxic T cells which may curtail the proliferation of tumor cells through two mechanisms: macrophages and interferon (IFN)-γ, activation [140]. Consistent with the ability of PC to evade tumor immunoediting, CD8^+^ T cells are rarely detectable in PC and their presence is associated with better prognosis (Figure 1) [141]. During tumor progression, PC evades CD8^+^ cytotoxicity, by causing both direct CD8^+^ aggregation in the desmoplastic tissue and indirect CD8^+^ inactivation through TGF-β secretion [142]. TGF-β release by PC cells in turn inhibits the transcription of CTL cytotoxic genes: perforin, granzyme A, granzyme B, Fas ligand, and IFN-γ, in vitro; these data support the hypothesis that TGF-β targeting could restore cytotoxic response involved in in vivo tumor clearance [143].

##### MDSC

MDSC are a heterogeneous group of immature cells classified in polymorphonuclear granulocytic MDSC and mononuclear monocytic MDSC, although they lack specific surface markers [144]. MDSC are represented in both human peripheral blood and TME and are involved in tumor growth, invasion and angiogenesis, through their activity on immune cells: for example, they promote the switch from M1 to M2 macrophage phenotype, promote Treg expansion and suppress natural killer (NK), CD4^+^, and CD8^+^ cells [81,82]. Indeed, in PC the levels of intratumor MDSC correlate with circulating Treg, but not with infiltrating CD8^+^ [145,146]. Moreover, circulating MDSC levels in PC patients are higher than in control subjects and increase upon disease progression [80]. Due to their contribution in promoting an immunosuppressive TME, blockade of the interactions between cancer cells and MDSC could improve clinical outcome by reactivating CD8^+^ cytotoxic activities. Baine and coworkers indeed demonstrated that the inhibition of tumor-derived granulocyte-macrophage colony-stimulating factor (GM-CSF, a cytokine highly expressed in PC and involved in disrupting antigen-specific T cells) prevents MDSC recruitment and tumor development [147].

##### DC

Antigen-presenting DC cells exert their effects in antitumor response mainly through MHC-mediated activation of CD4^+^ and CD8^+^ T cells [148]. Due to DC role in patrolling tissues to respond to danger stimuli, cancer cells develop strategies to evade immunosurveillance by blocking DC maturation and function; indeed, DC are rarely detected in PC TME [149,150]. Levels of both circulating and infiltrating DC correlates with better prognosis in both resectable and unresectable PC; furthermore, Fukunaka and coworkers reported a positive correlation between DC and CD4^+^/CD8^+^ tumor infiltration and an improved patient prognosis [132,151,152]. DC-based vaccination has been shown to gemcitabine’s therapeutic efficacy in the Panc02 poorly immunogenic murine tumor model [153]. Although preliminary clinical results with DC-based vaccination have shown tolerability and immunogenicity, clinical efficacy remains to be demonstrated [154,155].

### 3.2. Acellular Components

#### 3.2.1. ECM

PC is histologically characterized by the abundance of ECM which, in addition to mechanical support, influences both the cellular and molecular composition of pancreatic stroma [156]. This three-dimensional acellular portion is composed by fibrous proteins (e.g., collagen, fibronectin, laminin, proteoglycans (e.g., decorin, biglycan, and lumican)) and water, and represents a dynamic structure in continuous remodeling, especially during tumor progression [157]. In normal pancreatic tissue, PSC regulate ECM turnover through the synthesis and metalloproteinase (MMP)-2/MMP-9-dependent degradation of ECM proteins [158,159]. Since PSC activation is an early event during the neoplastic transformation, it is not surprising that PSC and PC cells bidirectionally influence ECM dynamics to sustain cancer progression: indeed, aPSC enhance ECM proteins production and remodeling, hence providing a more favorable microenvironment for tumor cell growth [160]. The excessive PSC-dependent ECM proteins deposition also contributes to increase tissue hypoxia, which could explain the biological mechanism behind the failure of antiangiogenic therapies in PC [119].

##### Collagens

Collagens are one the best characterized and most abundant components of PC’s ECM and 28 types with at least a triple-helical domain are currently identified [161]. In TME, PC cells intensively interact with interstitial collagens and these connections significantly foster the pro-tumorigenic features, such as EMT and invasion, through both integrin-dependent and -independent signaling [162]. Indeed, collagens can either bind integrin receptors or the tyrosine kinase discoidin receptor 1 (DDR1) expressed by PC cells. The combination of different α and β subunit heterodimers allows integrin receptors to bind different types of collagens, in order to activate several intracellular signaling pathways. The canonical integrin-dependent biological mechanisms comprise focal adhesion kinase-mediated cell migration and EMT through downregulation of E-cadherin [83,84]. On the other hand, binding of collagen to DDR1 promotes invadopodia formation and MMP-2/MMP-9 expression [163,164]. Recent evidence has demonstrated that different collagen types can play contrasting roles in tumor. Collagens I, IV and V are involved in tumor progression by affecting adhesion, migration and cell viability and their levels increase and correlate with cancer stage and decreased patients’ survival. Ohlund and collaborators demonstrated that collagen IV colocalizes with integrin receptors on cancer cell surface and their interaction maintains cell proliferation and avoids apoptosis [85]. Similar effects are exerted by collagen V, which has been identified as a tumor promoter in PC: indeed, Berchtold and coworkers showed that PSC release collagen type V, which promotes invasion and proliferation of PC cells [86]. On the other hand, collagen XV decreases during PC progression, consistent with its role in hindering protumor features [165]. These controversial roles played by different types of collagens reflect the plasticity of ECM: indeed, the collagens architecture is continuously modified during cancer progression in order to favor the acquisition of a more aggressive behavior (Figure 1). In that respect, collagens mainly contribute to provide mechanical cues and ECM stiffness, which increase resistance to conventional treatments [87].

##### Proteoglycans

Many of the ECM-characterizing proteins are glycoproteins, a large group of molecules divided in four families, according to cellular and subcellular localization, genetic homology and protein activity [166]. Proteoglycans are heavily glycosylated proteins and their pattern of N- or O- glycosylation fluctuates during neoplastic transformation, in order to promote cell proliferation, angiogenesis and immune response [167]. Among proteoglycans, decorin, biglycan, and lumican are all small leucine-rich proteoglycan and the most important in PC [167]. Decorin acts as a monomer Zn^2+^ metalloprotein and is the first proteoglycan shown to be involved in the regulation of cell growth, by inhibiting TGF-β, ERBB2 dimerization and receptor tyrosine kinase signaling [168,169,170,171,172]. Since it mainly displays tumor suppressive properties, it is not surprising that low levels of its expression are detected in the stroma of many solid tumors, including PC, and its production is suppressed at a transcriptional level by TNF-α [166,173]. Köninger and coworker demonstrated that decorin is mostly released by PSC rather than by tumor cells in PC; however, cancer cells are sensitive to the decorin-dependent anti-tumor activity of chemotherapeutic drug carboplatin [174]. Biglycan shares > 65% of homology with decorin, and consistently also biglycan is able to bind and inactivate TGF-β [175]. More specifically, Weber and his collaborators showed that biglycan is expressed by fibroblasts in PC and overexpressed by cancer cells in response to TGF-β. Moreover, as a compensative host defense mechanism, biglycan affects cell growth in a TGF-β-independent manner, by inducing the cyclin-dependent kinase inhibitor p27 and a consequent cell cycle arrest [175]. Yamamoto and coworkers investigated the role of lumican in PC progression and demonstrated that lumican-transfected PANC-1 cells display a hyperactivated ERK pathway, which stimulates cell growth and lowers invasive ability [88]. Li and colleagues correlated the expression of lumican with prognosis in 131 untreated PC patients undergoing surgery and observed that patients with higher levels of this proteoglycan had significantly increased OS after tumor resection, as compared to those with low lumican expression levels (*p* = 0.0006). In the same paper, they also demonstrated that lumican increases apoptotic signals via hypoxia-inducible factor (HIF)1-α blockade in PC cell lines in vitro [176]. Hence, the role of lumican in PC is not yet well clarified, probably due to differential glycosylation pattern of lumican between the experimental conditions of different analyses [88].

##### Fibronectin

Fibronectin is the most abundant acellular component within PC tumor stroma and is mainly produced by aPSC [159,177,178]. Fibronectin is recognized by integrins family members which represent the best characterized fibronectin receptors; through this binding, different processes such as proliferation, migration, adhesion and survival, are modulated [178]. An anti-β1 integrin antibody profoundly modifies pancreatic TME, hence counteracting fibronectin polymerization and fibrillar collagen I formation in PANC 1/3 T3 spheroids [179]. Moreover, Cortes and his colleagues demonstrated that tamoxifen treatment reduces the hypoxic response of PSC, leading not only to a reduction of fibronectin alignment and thickness, which are associated with lower cellular invasion, but also to an increase of fibronectin protein expression in in vitro experiments [180]. In PC, fibronectin production is related to the activation of the NF-κB pathway: Jagadeeshan and coworkers demonstrated that p21-activated kinase 1, a serine/threonine kinase significantly upregulated in PC as compared to normal tissue, stimulates fibronectin production by promoting the direct NF-κB binding to fibronectin promoter [181]. Fibronectin upregulation in PC is associated with advanced clinicopathological stage and tumor aggressiveness [177]. Moreover, fibronectin-induced EMT correlates with poor OS in surgically resected PC patients [89]. Currently, the use of multi-gene expression profile datasets and meta-analyses has led to the identification of fibronectin as one of ten hubs that may be involved in PC progression and pathogenesis [182]. Along these lines, focal adhesion kinase (FAK) activation downstream of integrin binding to fibronectin and collagens correlates with high levels of fibrosis and poor CD^+^ cytotoxic T cell infiltration in PC; FAK inhibition limited tumor progression, markedly reduced tumor fibrosis, decreased the numbers of tumor-infiltrating immunosuppressive cells, and rendered the previously unresponsive KPC mouse model responsive to T cell immunotherapy and programmed cell death protein (PD)-1 antagonists. [183].

#### 3.2.2. Cytokines

Cytokines are a family of soluble factors, classified into growth factors, chemokines, angiogenic factors and interferons and involved in tumor immune-landscape, by exerting pro- (IL-1β, IL-6, IL-8, and macrophage migration inhibitory factor—MIF) and anti-inflammatory (IL-10 and TGF-β) functions [184]. In TME, several cell types (e.g., immune, stromal, and tumor cells) release cytokines, involved in pleiotropic biological mechanisms through the binding to their specific cognate receptors in an autocrine and paracrine fashion. More in detail, cytokines affect the dynamic and complex immune network in TME, promoting PC immune-escape, PC progression, and drug response [184]. Evidence demonstrates that circulating pro-inflammatory cytokines are elevated in PC patients, as compared to healthy subjects, induce the PC cachexia clinical syndrome, and associate with clinical outcome, hence rendering these soluble factors new potential prognostic biomarkers and therapeutic targets (Figure 1) [185,186]. However, the role played by cytokines is often controversial, as the same factor (i.e., TNF-α) can be both a tumor promoter or a suppressor; thus, a recent systematic review suggested that considering a set of cytokines, rather than a single one, could be a more exhaustive and comprehensive approach [187]. Here, we summarize the biological implications of the most investigated cytokines potentially involved in PC disease course and prognosis.

##### IL-1β

IL-1β is a pro-inflammatory cytokine synthetized by monocytes, tissue macrophages, and PC cells; through the binding to the type 1 IL-1R, shared with IL-1α, the cleaved form of IL-1β activates NF-κB in PC cells [188]. Etoposide-resistant PC cell lines display elevated IL-1β production and NF-κB activity; blocking IL-1β signaling with an anti-IL-1R(I) antibody results in decreased NF-κB activity and restored chemosensitivity. Consistently, IL-1β produced by resistant A818-4 and PancTu-1 cell lines confers chemoresistance to the otherwise chemosensitive PT45-P1 cells, by enhancing the NF-κB [189]. IL-1β is also involved in promoting tumor aggressiveness by directly modulating the migration ability of both inflammatory and tumor cells through the regulation of the integrin superfamily. Indeed, IL-1β reduced integrin signaling in PC cell lines via JNK activation: JNK silencing counteracts IL-1β mediated vinculin and α5-integrin blockade [190]. Moreover, Schmid and his colleagues demonstrated that the recruitment and adhesion of myeloid cells is increased by IL-1β-induced integrin α4β1 activity [90]. In PC patients, IL-1β levels correlated with poor progression free survival (PFS; *p* = 0.056), especially when combined with high IL-6 levels (*p* < 0.001; see also below) [91].

##### IL-6

IL-6 is another proinflammatory cytokine, which activates JAK 2 and signal transducers and activators of transcription (STAT)1 and STAT3, through the IL-6 presentation to the signal-transducer gp130 by IL-6 receptor [191]. Once activated, JAK stimulates signaling through downstream pathways (MAPK, PI3K and STAT) involved in the promotion of cell cycle progression and proliferation. Furthermore, IL-6 is involved in PSC-mediated EMT in PC cells, as demonstrated by the reversion of STAT3-dependent EMT after either anti-IL-6 antibody treatment or STAT3 inhibition [92]. IL-6 may additionally contribute to the regulation of angiogenesis and metastatization, through the modulation of multiple soluble players. Indeed, IL-6 promotes the expression of angiogenesis-related genes such as VEGF, neuropilin (NRP)-1 and IL-10 [93]. Moreover, IL-6, IL-10, TGF-β, and VEGF together orchestrate the inhibition of dendropoiesis, thus shaping an immunosuppressive TME [94]. In PC patients, IL-6 levels are higher in metastatic disease than in locally advanced tumors (*p* = 0.0001) [94] and IL-6 released by myeloid cells in TME activates STAT3 signaling in PC cells, hence affecting poor patient’s outcome [192].

##### IL-8

Initially identified as a neutrophil-activating peptide, IL-8 is now a well-established proinflammatory chemokine, which signals through the binding to the CXCR1/2 receptors, thereby activating several G-protein-mediated signaling cascades involved in conferring several pro-tumoral properties to cancer cells [193]. Indeed, due to the presence of their receptors in several TME cell types (e.g., neutrophils, macrophages and endothelial cells), IL-8 regulates a plethora of mechanisms, such as cell proliferation and migration, angiogenesis, EMT and immune-escape [95]. PC cell lines release higher levels of IL-8 in culture medium, as compared to non-transformed HPDE cells; moreover, exogenous IL-8 increases the expression of the pro-angiogenic factor VEGF and its receptor NRP-2 and the levels of ERK phosphorylation, directly promoting tumor angiogenesis and cell proliferation, respectively [96]. Another group described a model in which stromal and tumor cells influence each other to enhance invasion and angiogenesis: in co-culture conditions, cancer cells stimulate fibroblasts to release higher levels of CXCL12 which, in turn, upregulates PC cell-derived IL-8 [194]. IL-8 is also involved in promoting ECM remodeling through MMP-2 and MMP-9 production [195]. The IL-8/CXCR1 axis is associated with the acquisition of CSC markers, such as CD44 and CD133, and functional abilities and correlates with lymph-node metastasis (LNM) (*p* = 0.012) and shorter OS in PC patients (*p* < 0.001) [97]. Furthermore, Chen and colleagues showed that PC patients express significantly higher IL-8 levels as compared to those with pancreatitis, in both tumor tissues and blood [196].

##### MIF

Another pleiotropic pro-inflammatory cytokine often overexpressed in PC is MIF. Although MIF was initially identified as a specific product of activated T lymphocytes, it is currently known that macrophages represent the main source of this soluble factor: TNF-α activation regulates MIF production and viceversa in an auto-amplification loop [197]. MIF interacts with the CD74 receptor, although complete activation requires the recruitment of different co-receptors, such as CD44 or CXCRs (e.g., CXCR2, CXCR4) [198]. MIF exerts its oncogenic role in PC through the downregulation of E-cadherin and the stimulation of vimentin and Zinc finger E-box-binding homeobox 1/2 production, hence causing EMT. Moreover, MIF-promoted PC aggressiveness is associated with increased resistance to gemcitabine treatment [98]. Recently, the detection of higher levels of MIF in PC tissue as compared to normal tissue increased the scientific interest in studying MIF as a new potential therapeutic target: MIF-overexpressing PC are more aggressive and result in poor OS [98,99]. Consistently, in vitro experiments demonstrated that the MIF post-transcriptional silencing could lead to PC regression through the induction of apoptosis and cell cycle arrest [199].

##### IL-10

Monocytes, macrophages, Th2 CD4^+^ T cells and activated B cells are the main source of IL-10 production in TME. The IL-10R1 is involved in the activation of specific IL-10 functions and the heterodimerization of IL-10R1 and IL-10R2 chains is required for outside-in cell signaling [200]. As an inhibitor of inflammatory responses, IL-10 counteracts the productions of proinflammatory cytokines (e.g., IFN-γ, IL-1β, and IL-6) from different cell types and DC maturation by blocking IL-12 expression [201]. IL-10 inhibits CD4^+^ T-cell response induced by mesothelin, an immunogen involved in promoting metastasis and tumor proliferation and a potential therapeutic target in PC [100]. Consistently, blocking IL-10 partially restores mesothelin-CAR T-cell activity [202]. IL-10 levels are increased in PC patients, as compared to healthy subjects, and correlate with poor survival of PC patients [101,102].

##### TGF-β

TGF-β plays an essential, but also complex and controversial, role in PC. TGF-β maintains homeostasis in normal tissue; however, being genetically unstable entities, cancer cells have the capacity to corrupt this suppressive influence. As a consequence, through the binding to the TGF-βRI and TGF-βRII heterotetrameric complexes, pathological forms of TGF-β signaling promote tumor growth and metastasis by activating downstream canonical Smad-dependent, as well as Smad-independent, intracellular pathways [203,204,205]. Activated phospho-Smad translocates into the nucleus, where it binds Smad-binding elements and transactivates TGF-β-dependent genes; non-Smad dependent signaling, on the other hand, mainly promotes activation of the MAPK pathway through RAS [206]. Regardless of the activated signaling, TGF-β acts as an EMT inducer [103]. Ellenrieder and collaborators demonstrated that MEK inhibition reduces TGF-β-dependent EMT and PC cells migration in *KRAS* mutant contexts [207]. TGF-β is also involved in modulating immune response against cancer cells: while DC stimulate CD8^+^ differentiation in co-culture with PC cells, TGF-β blockade improves antigen specific response and decreases T cells apoptosis; as a consequence, DC vaccination in combination with TGF-β blockade results in increased DFS in murine tumor models [104]. In PC, TGF-β levels correlate with clinical outcome and different groups demonstrated that PC patients with high levels of soluble TGF-β have a worse prognosis, as compared to those with low levels [105,106].

##### TNF-α

Similar to TGF-β, TNF-α plays a controversial role in PC stage progression and therapeutic response. Indeed, TNF-α exerts its activity as both a pro-tumoral and an anti-tumoral soluble factor [208]. This dual role is mainly due to the presence of two different receptors, TNFR1 and TNFR2, associated with pro-inflammatory and anti-inflammatory signaling, respectively [209]. As a type II transmembrane protein, TNF-α can bind both TNFR1 and TNFR2, whereas as the soluble form derived from proteolytic cleavage is able to activate TNFR1 exclusively [209,210]. While the ubiquitously expressed TNFR1 is mainly involved in apoptotic signaling, the role of TNFR2, mainly expressed on immune cells, is poorly understood [211]. PC cells-derived TNF-α enhances the tumor cell growth and invasiveness both in vitro and in vivo [107]. M1 macrophages also induce matrix remodeling through EMT in a TNF-α-dependent manner, highlighting the importance of TNF-α as a mediator in pancreatic TME [108]. The levels of TNF-α are higher in serum derived from PC patients, as compared to chronic pancreatitis or healthy patients [212]. However, gemcitabine treatment upregulates TNF-α production, resulting in a potent antitumor activity [213]. TNF-α-dependent anti-tumoral activity is limited by NF-κB activation, which can be overcome by the combination between gene therapy through an adenoviral vector-expressing TNF-α and nafamostat mesilate, a NF-κB inhibitor which upregulates apoptosis both in in vitro and in vivo [214].

## 4. Current Treatment Standard of Care and Novel Therapeutic Strategies

The overall five-year survival rate for PC patients remains below 10% (all stages included) and although median OS in advanced disease has improved with modern chemotherapy it remains below one year [2]. Approximately 80% of PC patients presents with surgically unresectable or metastatic cancer at diagnosis and among resected patients most will recur with metastatic disease within 3 years from surgery. Standard treatment strategies usually become rapidly ineffective, due to the early development of drug resistance; the cornerstone of PC treatment is conventional cytotoxic chemotherapy, and no predictive biomarkers are currently available to select patients for targeted anti-cancer treatments or immunotherapy. Thus, novel therapeutic approaches are urgently needed to improve the dismal prognosis of PC patients.

So far, results from clinical trials testing agents targeting cell autonomous signaling aberrations and angiogenesis have been quite disappointing in PC. Clinical trials investigating inhibitors of the EGFR family of receptors, of the MAPK and of the Hedgehog signaling pathways have failed to demonstrate an improvement in OS in PC patients [215,216,217]. Similarly, targeting the VEGF/VEGFR axis alone or in combination with an anti-EGFR, have also met with little clinical success; nevertheless, a small randomized phase II trial did demonstrate potential advantages for the multikinase angiogenesis inhibitor sunitinib in the first-line maintenance setting [218,219].

So far, alterations in the DNA damage repair machinery appear to be the most promising target(s) in PC [220]. Five to 15% of PC are characterized by either somatic or germline *BRCA* mutations and several case studies reported responses to poly-ADP-ribose-polymerase (PARP) inhibitors [221,222]. While a phase II trial of veliparib failed to demonstrate increased response in PC patients previously treated with chemotherapy, maintenance therapy with olaparib after a platinum-containing 1st line regimen prolonged OS in patients with non-progressing PC patients with germline BRCA1/2 mutations in the phase III trial POLO; olaparib is thus the first PARP inhibitor being granted approval by United States Food and Drug Administration and European Medicines Agency for the treatment of advanced PC [49,223]. Additional trials are ongoing to evaluate combination therapy with PARP inhibitors and chemotherapy or other targeted therapy (NCT01489865, NCT0158805). Similarly, a variable proportion of PC patients, whose tumors are characterized by defective mismatch DNA repair machinery and microsatellite instability (MSI-H), appears to experience prolonged responses to immune checkpoint inhibition by anti-PD1 antibodies [224].

More recently, attempts at modulating TSI with agents targeting different TME components, have also failed: pegvorhyaluronidase alfa, an agent targeting tumor hyaluronan, napabucasin, an agent targeting cancer cell stemness, recombinant pegylated IL-10, and a combination of anti-CTLA-4 and anti-PD-L1 antibodies have all recently shown lack of efficacy in clinical trials in advanced PC patients.

The largest effort of integrated genomic analysis in understanding the molecular pathology of PC has recently confirmed the TGF-β as the most recurrently mutated signal transduction pathway in PC [63]. The therapeutic efficacy of the TGF-β canonical Smad-dependent pathway inhibition by using the small molecule selective inhibitor of the TGF-βRI LY2109761 (Lilly, Indianapolis, USA) was initially demonstrated in preclinical models of PC [225]. Moreover, the role of a unique Smad-independent TGF-β pathway represented by the TGF-β activated kinase 1 (TAK1) was dissected in PC [226]. This serine/threonine kinase has a critical role in sustaining resistance to chemo- and radiotherapy in PC, by integrating signals from various cytokines and controlling, in turn, the activation of different transcription factors, including NF-κB [227,228,229]. More recently, a fundamental role of TAK1 was demonstrated towards PC chemoresistance through the kinase-independent modulation of two transcriptional regulators emerging as central determinants of malignancy in this disease, namely the transcriptional regulator yes associated protein (YAP) and the transcriptional coactivator with a PDZ-binding domain (TAZ) [230]. In light of these observations, TGF-β signaling represents a key target for personalized treatment of PC. Different therapeutic strategies have been developed and demonstrated efficacy in several preclinical and clinical trials in PC. Among the approaches used to target TGF-β, it is worth to mention trabedersen (AP12009), a TGF-βII antisense RNA molecule, which inhibits proliferation and migration of human PC cells and reduces growth and metastasis in human PC orthotopic mouse models [231]. In addition, a trabedersen phase I dose-escalation trial is ongoing in advanced PC patients (NCT00844064). Soluble TGF-βII was also used as another therapeutic strategy to inhibit of TGF-β binding to cellular TGF-βR in human PC cells, thereby suppressing tumor growth, angiogenesis and migratory capability of PC [232,233]. In addition, a phase I/II trial is ongoing in advanced solid tumors, including PC, with the anti- TGF-β antibody NIS793, as single agent or in combination with anti-PD-L1 (NCT02947165). Among TGF-β inhibitors acting at the intracellular signaling level, inhibitors of TGF-βRI kinase (i.e., SD208, SD093 and LY580276) significantly reduced the aggressiveness of PC cells in preclinical models. In particular, LY580276 is an ATP-competitive inhibitor of the TGF-βRI/ALK5 shown to arrest EMT and migration in PC cells [234,235]. A multi-center, open-label, phase Ib clinical trial is ongoing to evaluate the safety, tolerability, and exploratory efficacy of vactosertib (TEW-7197)—a small molecule inhibitor of TGF-βRI kinase—in combination with FOLFOX second-line chemotherapy in patients with metastatic PC after first-line gemcitabine and nab-paclitaxel (MP-PDAC-01, NCT03666832 [236]. The inhibitor of TGF-βRI/II kinase LY2109761 demonstrated to reduce tumor growth, metastasis and survival in combination with chemotherapy in an orthotopic murine model of advanced PC [225]. Despite the great number of TGF-βR inhibitors developed, only a few of them progressed to clinical investigations mainly due to relevant cardiac adverse events in animals. Compared to others small molecule inhibitors, galunisertib (LY2157299 monohydrate), an oral small molecule inhibitor of the TGF-βRI kinase that specifically downregulates the phosphorylation of Smad2, has demonstrated a reduced cardiovascular toxicity profile in preclinical models and it was the first-in-class to start its clinical development in PC patients [237]. The phase Ib/randomized phase II, double-blind trial in patients with advanced or unresectable PC, H9H-MC-JBAJ, demonstrated that the combination of galunisertib and gemcitabine improved OS versus gemcitabine monotherapy with an expected manageable toxicity [238]. Based on these positive results, the phase I clinical trial I8X-MC-JECA has been designed and is ongoing in PC patients, to test the safety of the next generation small TGF-βRI inhibitor LY3200882, in combination with the most recent standard first-line chemotherapy, gemcitabine and nab-paclitaxel (NCT02937272). Translational studies have been conducted to measure plasma proteins and miRNA markers at baseline and during treatment in patients enrolled in the H9H-MC-JBAJ study [239]. Predictive markers, useful to select patients who may benefit more from TGF-βR inhibition, have been identified. In particular, macrophage inflammatory protein (MIP)-1-α (a proinflammatory cytokine involved in the recruitment of monocytes and macrophages) and INFγ-induced protein (IP)-10 (involved in the recruitment of Treg) were two of the four most significant predictive biomarkers for chemoresistance in the population receiving single agent gemcitabine (for MIP-1-α mOS 95% CI high vs. low, 3.6 (2.5–4.0) vs. 10.1 (7.2–17.0) months, HR 95% CI = 3.93 (2.13–7.22) *p* < 0.01; for IP-10 mOS 95% CI high vs. low, 3.6 (2.5–7.7) vs. 12.5 (7.1–17.0) months, HR 95% CI = 2.78 (1.55–4.99) *p* < 0.001) [240]. On the contrary, treatment with galunisertib demonstrated to dramatically revert this resistance as MIP-1-α and IP-10 were the most significant positive predictive biomarkers for the combination of this agent with chemotherapy (for MIP-1-α mOS 95%CI galunisertib + gemcitabine vs. gemcitabine, 9.2 (5.5–12.4) vs. 3.6 (2.5–4.0) months, HR 95% CI = 0.38 (0.22–0.66) *p* < 0.01; for IP-10, 10.9 (5.5–12.4) vs. 3.6 (2.5–4.0) months, HR 95% CI = 0.4 (0.23–0.67) *p* < 0.0015) [239]. These results suggested an involvement of the immune system in the response of PC to galunisertib. Unfortunately, results from clinical trials using cancer vaccines or immune-checkpoint inhibitors (anti-CTLA4 and anti-PD-L1) in PC have been disappointing so far, thus immunotherapy remains a therapeutic challenge in PC patients [241,242,243,244]. The challenge for these approaches resides in the fact that, unlike melanoma, PC is characterized by a highly immune-suppressive microenvironment with a very low number of effector T cells. For this reason, research is moving towards the use of microenvironment re-educators, such as TGF-β inhibitors, in combination with checkpoint inhibitors, as a strategy for “immune-excluded” tumors, aiming to converting a non-immunologic tumor into an immune-responsive disease. Preclinical data from PC mouse models demonstrated that silencing of TGF-βRI induced a recruitment of CD8^+^ lymphocytes around PC [245]. Based on all these results, it has been conducted a phase Ib dose-escalation and cohort-expansion clinical trial of galunisertib in combination with the anti-PD-L1 antibody durvalumab in refractory metastatic PC patients (H9H-MC-JBEG). In the 32 patients treated with the recommended phase II dose, one partial response and 7 stable diseases were observed (disease control rate 25%); PFS was 1.9 months (95% CI: 1.5, 2.2) and mOS was longer than 12 months (95% CI: 3.6, NR). This combination has demonstrated to be active and well-tolerated [246]. Although these results represent the first signal of activity for an immune checkpoint-inhibitor combination strategy in an “immune-excluded” tumor as PC, its efficacy would need to be confirmed in larger randomized trials in advanced PC patients. Nonetheless, having demonstrated that TGF-β signaling inhibition is an active strategy in combination with classic chemotherapy and immune checkpoint-inhibitors in unselected advanced PC patient populations, the identification of biomarkers that could predict which patients would benefit more from these combination strategies remains of unique importance for the future development of more personalized treatments.

## 5. Conclusions

PC remains arguably the deadliest of solid tumors. Advances in the understanding of the genomic and transcriptomic complexity and heterogeneity of this disease, as well as elucidation of basic biological mechanisms underlying PC transformation and progression to a highly malignant phenotype, are of paramount importance to drive further clinical research and improve survival. However, clinical experiments clearly show that, with few exceptions, targeting individual players within the extremely complex and multifaceted crosstalk between the cancer cell genetic background, soluble factors, and cellular and acellular components of a usually highly desmoplastic stroma represents an oversimplification endowed with little potential for clinical efficacy.

Research efforts should thus concentrate on dissecting the currently elusive Ariadne’s string that links early occurring genetic aberrations of transformed epithelial cells, with their ability to secrete specific soluble factors (growth factors, cytokines, chemokines) which, in turn, will shape the molecular and cellular composition of the TME; TME, on the other hand, will respond to tumor growth (and presumably to the selective pressure applied by treatments) by activating different sets of soluble factors which, in turn, will influence tumor cell behavior and reshape the TME. In depth understanding examples of how the tumor genetic background can influence TME composition and response to treatments include, for example, evidence that loss of the PTEN tumor suppressor results in immunotherapy resistance, by directly inducing the T-cell inhibitor PD-L1 in several malignancies, including in mouse pancreatic cancer models [247,248,249]. Another mechanism by which tumor genotype can directly affect TME composition is the expression of specific cytokines, often involved in characterizing an immunosuppressive environment [250]. In that respect, it has been demonstrated that the oncogenic *KRAS* mutations are associated to an increased production of the GM-CSF, which causes the expansion of immunosuppressive Gr1^+^CD11b^+^ myeloid cells in mouse PC models; GM-CSF inhibition, in turn, could prevent MDSC recruitment and tumor development [251].

TME, on the other hand, may activate tumor cell signaling pathways which, in turn, cause profound TME remodeling: for example, FAK by PC cell’s integrin binding to fibronectin and collagens induces fibrosis and poor CD8^+^ cytotoxic T cell infiltration; inhibition of intracellular FAK signaling reduces tumor fibrosis, decreases the numbers of tumor-infiltrating immunosuppressive cells, and restores PC sensitivity T cell immunotherapy and PD-1 antagonists [183].

Cytokines secreted by either PC cells or stromal elements in the TME have pleiotropic, and sometimes unpredictable, effects: for example, both CAF and aPSC act in an autocrine and paracrine fashion in order to modulate the activity of the other cellular types in TME, such as endothelial and immune cells [252]. Cancer cells release proangiogenic soluble factors to induce neovascular formation in hypoxic condition of growth; on the other hand, hypoxia is one of the external stimuli of PSC activation and induces aPSC to secrete proangiogenic elements, which not only promote endothelial cells proliferation, but also sustain the maintenance of aPSC features [253].

Finally, TME components (both soluble factors and cellular and structural components, often display a double-edged function: for example, Tregs usually create an immunosuppressive, therapy resistant environment, but their depletion in PC may actually promotes pancreatic carcinogenesis, as a result of CAF activation and immunosuppressive myeloid infiltration. As another example, IL-10 function as an anti-inflammatory mediator, counteracting the production and activity of pro-inflammatory cytokines such as IL-1β and IL-6, which have been shown to favor PC progression and aggressiveness, has led to its development as a therapeutic agent; however, its double-edged activity as a suppressor of CD4 responses and DC maturation may possibly explain its therapeutic failure in recent clinical trials in PC.

It affords from the above that finding such elusive Ariadne’s string will be of paramount importance in moving PC therapeutic scenario forward, avoiding a dreadful waste of time and patients’ lives within dead-end paths. In addition to better understanding biology, we will have to improve our clinical methodology, by designing smart and effective clinical trials to: (1) provide clinical proof of concept supporting the proposed biological mechanism; (2) identify relevant biomarkers; (3) lay solid foundations for success in phase III clinical trials.

## Figures and Tables

**Figure 1 cells-09-00309-f001:**
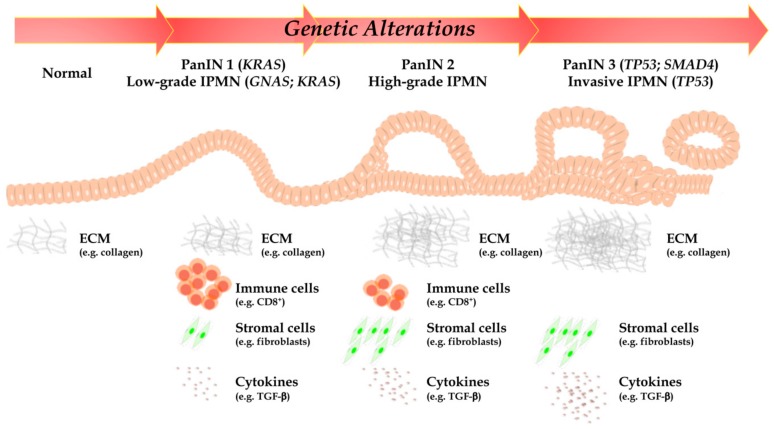
According to PanIN and IPMN stage, a time-manner dependent accumulation of point mutations in oncogenes and tumor suppressor genes, epigenetic alterations and chromosomal structural variants are represented. Furthermore, PC is mainly represented by desmoplastic stroma and immunosuppressive TME: indeed, cell populations, collagen organization, and cytokines are profoundly different between normal pancreatic tissue and advanced PC. DC, dendritic cells; *GNAS*, guanine nucleotide binding protein, alpha stimulating; IPMN, intraductal papillary mucinous neoplasm; *KRAS*, Kirsten rat sarcoma; MDSC, myeloid-derived suppressor cells; PanIN, pancreatic intraepithelial lesion; PC, pancreatic cancer; PDAC, pancreatic ductal adenocarcinoma; PSC, pancreatic stellate cells; TME, tumor microenvironment; *TP53*, tumor suppressor protein 53.

**Table 1 cells-09-00309-t001:** The most important genetic pathways altered in PC.

Name of the Pathway	Most Important Mutated Gene (s)	
	Oncogenes	Tumor Suppressor
MAPK signaling	*KRAS*	
DNA damage control		*TP53*, *BRCA1*, *BRCA2*
Control of G1/S Phase transition		*CDKN2A*
TGF-β signaling	*TGF-βRII*	*SMAD4*
Apoptosis		*CASP10*, *VCP*
Hedgehog signaling	*SOX3*, *GLI1*, *GLI3*	
Homophilic cell adhesion		*CDH1*, *FAT*
Integrin signaling	*ITGA4*, *ITGA9*, *LAMA1*, *LAMA4*, *LAMA5*	
JNK signaling	*MAP4K3*, *TNF*	
SWI/SNF		*ARID1A*
Small GTPase signaling	*RP1*	
WNT/Notch signaling	*MYC*	*TSC2*
Axon guidance		*SLIT2*, *ROBO2*

*ARID1A*, AT-rich interaction domain 1A; *CDKN2A*, cyclin dependent kinase inhibitor 2A; GTP, guanosine-5’-triphosphate; *ITGA*, integrin alpha subunits; *KRAS*, Kirsten rat sarcoma; *LAMA*, laminin subunit alpha 1; *MAP4K3*, mitogen-activated protein kinase kinase kinase 3; *SOX*, sry-related HMG box; SWI/SNF, switch/sucrose non-fermentable; *TGF*, transforming growth factor; *TNF*, tumor necrosis factor; *TP53*, tumor protein 53; *TSC*, tuberous sclerosis complex; *VCP*, valosin containing protein.

**Table 2 cells-09-00309-t002:** Ongoing clinical trials investigating signaling pathway inhibitors in patients with PC.

Study Drug	Sponsor	Treatment Setting	Combination Partner	Study Phase	Clinical Trials.gov ID
**ERK1/2 Inhibitors**					
Ulixertinib	University of Washington School of Medicine	First Line	Nab-paclitaxel plus gemcitabine	I	NCT02608229
	UNC Lineberger Comprehensive Cancer Center	Second Line	Palbociclib	I	NCT03454035
KO-947	Kura Oncology	Second Line	-	I	NCT03051035
**CDK 4/6 Inhibitors**					
PD-0332991	Dana Farber Institute	Any	Gedatolisib	I	NCT03065062
Palbociclib	Pfizer	Any	Nab-paclitaxel	I	NCT02501902
**KRASG12C Inhibitors**					
AMG 510	Amgen	Any	-	I	NCT03600883
**ALK Inhibitors**					
Ceritinib	Roswell Park Cancer Institute	Any	Gemcitabine ± nab-paclitaxel or cisplatin	I	NCT02227940
**RTK Inhibitors**					
Entrectinib	Hoffmann-La Roche	Any	-	II	NCT02568267
Larotrectinib	Loxo Oncology	Second Line	-	II	NCT02576431

ALK, anaplastic lymphoma kinase; CDK, cyclin-dependent kinase; ERK, extracellular signal–regulated kinase; KRAS, Kirsten rat sarcoma; RTK, receptor tyrosine kinase.

**Table 3 cells-09-00309-t003:** Contribution of protumor TME components in PC.

TME Components			Biological Mechanisms	Implication in PC	Ref.
Cellular	Stromal cells	aPSC	↑tumor proliferation, ECM production, EMT, proangiogenic soluble factors secretion; ↓CD8^+^ T cells migration	↑chemoresistance; ↑antitumor immune response	[67,68,69,70]
		CAF	↑tumor proliferation, production of proangiogenic soluble factors, M2 polarization	↑chemoresistance; ↓OS	[71,72,73]
		Endothelial	↑antitumor immune cells	↑OS	[74]
	Immune cells	M2	↑Snail, vimentin and EMT; ↓E-cadherin	↓patient prognosis	[75,76]
		Neutrophils	↓immunoresponse	↑chemoresistance	[77]
		CD4^+^/CD25^+^/Foxp3^+^ Treg	↑immunosuppressive TME; ↓antitumor immune cells	↑TNM stage	[78,79]
		MDSC	↑immunosuppressive TME	↑tumor growth invasion and angiogenesis	[80,81,82]
Acellular	ECM	Collagen I, IV, V	↑tumor proliferation and EMT	↓patient prognosis	[83,84,85,86,87]
		Lumican	↑tumor proliferation	↓patient prognosis	[88]
		Fibronectin	↑EMT	↓patient prognosis and OS; ↑chemoresistance	[69,89]
	Cytokines	IL-1β	↑inflammation, migration	↑chemoresistance; ↓PFS	[90,91]
		IL-6	↑tumor proliferation, angiogenesis, EMT and immunosuppressive TME	↓patient prognosis	[92,93,94]
		IL-8	↑tumor proliferation, angiogenesis, CSC properties, ECM disruption and migration	↓patient prognosis	[95,96,97]
		MIF	↑cell proliferation and EMT	↑chemoresistance; ↓OS	[98,99]
		IL-10	↓CD4^+^ T-cell response	↓OS	[100,101,102]
		TGF-β	↑EMT; ↓CD8^+^ T-cell apoptosis	↓OS	[103,104,105,106]
		TNF-α	↑tumor proliferation, EMT and migration	↓patient prognosis	[107,108]

aPSC, activated pancreatic stellate cells; CAF, cancer associated fibroblasts; CSC, cancer stem cells; DC, dendritic cells; ECM, extracellular matrix; EMT, epithelial mesenchymal transition; IL, interleukin; M2, macrophages M2 phenotype; MDSC, myeloid-derived suppressor cells; MIF, macrophage migration inhibitor factor; TGF, transforming growth factor; TNF, tumor necrosis factor; TNM, tumor node metastasis; OS, overall survival; PFS, progression free survival.

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
