# Peer review of "From Genetic Alterations to Tumor Microenvironment: The Ariadne’s String in Pancreatic Cancer"

_cells, 2020, doi:10.3390/cells9020309_

Round 1

Reviewer 1 Report

The manuscript “From genetic alterations to tumor microenvironment: the Ariadne’s string in pancreatic cancer” by Bazzichetto et al. is within the scope of the journal. The manuscript is well written and covers relevant and interesting topic, however, there are certain concerns which needs to be addressed before it can be considered for acceptance.

General comments

Please describe PanIN and IPMN in more detail. Also include IPMN in Figure 1. Please include genetic alterations instead of just writing genetic alterations. In Table 1. Please include a column stating whether the pathway is considered oncogene or tumor suppressor. Add sub headings to the driver genes alterations such as KRAS, TP53 etc. In Table 3 Please make sub category of immune cells. Why neutrophils are not included? M2 has not been defined. Also include endothelial cells in this table. Also, give subheadings in cellular, acellular and cytokines paragraph. This will make reading the review easier for a reader. In all the p value listed please include spaces (eg. p = 0.05 and not p=0.05)

Comments

The review lacks connection between genetic alterations and tumor microenvironment section. If author thinks that signaling pathways are common connections between the two, please state it clearly. The two sections need a connection to give this review a rationale and logic. The section on tumor microenvironment lacks flow. It seems like paragraphs with lots of information but lacks conclusion or rationale which author want to communicate and needs to be stated. Similarly, cytokine section needs a logical flow and conclusion. The author should define what they think is the ariadne’s string and what could be the possible solution. Conclusion needs more details.

Reviewer 2 Report

Pancreatic cancer (PC) is a lethal malignancy due to asymptomatic nature, early metastasis, and poor efficacy of the current therapeutic regimens. The mutations leading to constitutive activation of K-ras are the major initiating event followed by alterations in CDKN2A, p53, and SMAD4 genes. The development of the intense and complex desmoplasia is the characteristic feature of the PC, which results in the obstructive and immunosuppressive microenvironment. The extensive fibrosis and deposition of the extracellular matrix proteins (ECM) lead to poor perfusion and hypoxic conditions resulting in the evolution of the aggressive clones. The crosstalk between stroma and cancer cells, mediated by soluble factors, is critical for the aggressiveness and therapy resistance in PC. The manuscript submitted by Bazzichetto et al reviewed the current understanding of driver mutations, development of the intricate microenvironment and its role in the PC pathobiology. In addition, the authors describe the current status of the standard-of-care and ongoing clinical trials.

           Overall, the manuscript is well written. However, the majority of the aspects discussed are introductory, scattered or superficially described. It will be better if authors focused on either biology or novel therapeutic strategies in PC.    

Minor concerns.

Figure 1 does not accurately describe the PC progression, complexity, and the characteristic features as mentioned in legends. 1) It will be better to include the mutational profile along with PC progression; 2) Majority of PC have lower epithelial cellularity, 3) MDSC are absent or low during PanIN stage, 4) Fibroblast are more abundant during PanIN stage compared to normal pancreas, and 5) collagen is less in the normal pancreas (mainly present with inflammation and cancer). Pancreatic cancer is abbreviated as PC as well as PDAC in the manuscript. Choose one abbreviation.  Include the study by Jiang et al, Nature Medicine. 2016; 22(8):851-60 in the last section.    The following statements need to be rephrased for better understanding.  Line 160-161, what are infiltrating components? Line 241, abbreviate pancreatic cancer as PC. Line 300 “immunity disruption thereby causing chemoresistance.”  Line 362-363 “…T-cell mainly activate ……therefore, their percentage decreases in tumor tissues and peripheral blood”.  Line 442-443 need correction Line 479 and 480 correct TGF-β abbreviations. Line 671 correct the “filed”. Also check the lines 607, 615, 649, 657.  

Round 2

Reviewer 1 Report

The manuscript can be accepted.

Reviewer 2 Report

1. Discuss briefly the current standard-of-care for pancreatic cancer at the beginning of Section 4.